# Effect of Particulate Matter Exposure on Respiratory Health of e-Waste Workers at Agbogbloshie, Accra, Ghana

**DOI:** 10.3390/ijerph17093042

**Published:** 2020-04-27

**Authors:** Afua Asabea Amoabeng Nti, John Arko-Mensah, Paul K. Botwe, Duah Dwomoh, Lawrencia Kwarteng, Sylvia Akpene Takyi, Augustine Appah Acquah, Prudence Tettey, Niladri Basu, Stuart Batterman, Thomas G. Robins, Julius N. Fobil

**Affiliations:** 1Department of Biological, Environmental & Occupational Health Sciences, School of Public Health, University of Ghana, P.O. Box LG13 Accra, Ghana; jarko-mensah@ug.edu.gh (J.A.-M.); pkbotwe@ug.edu.gh (P.K.B.); lawkwarps@yahoo.com (L.K.); phykles@gmail.com (S.A.T.); nana.austine@gmail.com (A.A.A.); ptettey@ug.edu.gh (P.T.); jfobil@ug.edu.gh (J.N.F.); 2Department of Biostatistics, School of Public Health, University of Ghana, P.O. Box LG13 Accra, Ghana; duahdwomoh@gmail.com; 3Faculty of Agricultural and Environmental Sciences, McGill University, Montréal, QC H9X 3V9, Canada; niladri.basu@mcgill.ca; 4Department of Environmental Health Sciences, University of Michigan, 1415 Washington Heights, Ann Arbor, MI 48109-2029, USA; stuartb@umich.edu (S.B.); trobins@umich.edu (T.G.R.)

**Keywords:** PM exposure, breathing zone, informal e-waste workers, Agbogbloshie, lung function, longitudinal study

## Abstract

*Background*: Direct and continuous exposure to particulate matter (PM), especially in occupational settings is known to impact negatively on respiratory health and lung function. *Objective*: To determine the association between concentrations of PM (2.5, 2.5–10 and 10 µm) in breathing zone and lung function of informal e-waste workers at Agbogbloshie. *Methods*: To evaluate lung function responses to PM (2.5, 2.5–10 and 10 µm), we conducted a longitudinal cohort study with three repeated measures among 207 participants comprising 142 healthy e-waste workers from Agbogbloshie scrapyard and 65 control participants from Madina-Zongo in Accra, Ghana from 2017–2018. Lung function parameters (FVC, FEV1, FEV1/FVC, PEF, and FEF 25-75) and PM (2.5, 2.5–10 and 10 µm) concentrations were measured, corresponding to prevailing seasonal variations. Socio-demographic data, respiratory exposures and lifestyle habits were determined using questionnaires. Random effects models were then used to examine the effects of PM (2.5, 2.5–10 and 10 µm) on lung function. *Results*: The median concentrations of PM (2.5, 2.5–10 and 10 µm) were all consistently above the WHO ambient air standards across the study waves. Small effect estimates per IQR of PM (2.5, 2.5–10 and 10 µm) on lung function parameters were observed even after adjustment for potential confounders. However, a 10 µg increase in PM (2.5, 2.5–10 and 10 µm) was associated with decreases in PEF and FEF 25–75 by 13.3% % [β = −3.133; 95% CI: −0.243, −0.022) and 26.6% [β = −0.266; 95% CI: −0.437, 0.094]. E-waste burning and a history of asthma significantly predicted a decrease in PEF by 14.2% [β = −0.142; 95% CI: −0.278, −0.008) and FEV1 by 35.8% [β = −0.358; 95% CI: −0.590, 0.125] among e-waste burners. *Conclusions*: Direct exposure of e-waste workers to PM predisposes to decline in lung function and risk for small airway diseases such as asthma and COPD.

## 1. Introduction

Occupational exposure to particulate matter (PM) has been identified as a major contributing factor of respiratory diseases including reduced lung function. However, very few studies have examined the effects of exposure to PM on respiratory outcomes among informal electronic waste (e-waste) workers with none being conducted among e-waste workers in Ghana. There is the need to provide data to add to intellectual knowledge, inform public health policy and worker health education programming especially in Ghana.

Exposure to airborne particulate matter (PM) present in the work environment could precipitate a myriad of health conditions [1,2], including lung function changes, morbidity and mortality due to respiratory and cardiovascular diseases [3,4]. The association between exposure to ambient PM and reduced lung function parameters have been reported in many studies [5,6,7,8,9,10] and the degree and nature of effect is dependent on PM type, size, ambient concentration, duration of exposure and total lung ventilation period of exposed individuals [3,11]. Inhaled particles that can reach the lower airways are categorized into three fractions depending on the size; PM10 (diameter ≤10 μm), fine or PM2.5 (diameter ≤2.5 μm) and ultrafine, UFP or PM0.1 (diameter ≤1 μm). Due to their ubiquitous nature, fine and UFP particles easily penetrate deep into the lungs unfiltered, predisposing to various acute and chronic respiratory health effects. Inhalation of PM into the respiratory tract often induces oxidative stress and generation of reactive oxygen species production, resultant hypersensitivity and inflammation of respiratory lining of which may impact optimal lung function. Prolonged chronic exposure may cause lung tissue remodeling, scarring with fibrosis leading to chronic lung conditions such as asthma, bronchitis, chronic obstructive pulmonary disease (COPD), and even lung cancer [4,12].

One of the conundrums the world face is proper disposal and management of e-waste, due to the several toxic substances they contain. For example, informal electronic waste (e-waste) recycling activities such as manual dismantling and open air burning of cables release several toxic substances including polyromantic hydrocarbons, heavy metals and organic-bound PM into ambient environment [13]. Several studies conducted at the Agbogbloshie informal e-waste site have reported high concentrations of harmful compounds present in ambient air, soil sediments, water bodies and biological fluids (blood and urine) of e-waste workers [13,14,15]. For instance, (Caravanos, Clark, Fuller, & Lambertson, 2011) [16] measured high level of lead (0.98 mg/m^3^) in the breathing zone of one e-waste worker, which was twenty times the American Conference of governmental industrial hygienists’ value (ACGIH) permissible threshold limit value (TLV) of 0.05 mg/m^3^. Iron concentrations in the worker’s breathing zone ranged from 1.7–5.6 mg/m^3^, other heavy metals such as copper and aluminum were also above the accepted ACGIH TLV limits of 5.0 and 1.0 mg/m^3^ respectively.

The association between exposure to personal ambient PM and reduced lung function parameters has been reported in many occupational and epidemiological studies. These studies have shown a positive correlation between PM exposure and adverse respiratory health conditions [6,17,18,19]. For instance, both population-based and occupational studies have shown that short and long term exposure to PM, particularly, PM2.5 may predispose to bronchial asthma, chronic obstructive pulmonary disease, and increased respiratory symptoms or worsened pre-existing conditions in susceptible individuals [20]. However, few respiratory health effects studies have been conducted among informal e-waste workers. For instance, a survey among e-waste workers in India found 76% of the studied population reported respiratory illnesses such as dyspnea, coughs, asthma, and bronchitis [21]. The respiratory health effects of recurrent exposure to PM among informal e-waste recyclers have not been adequately studied. The aim of this study was to assess the effects of exposure to PM (2.5, 2.5–10 and 10 µm) in the breathing zone on lung function of healthy informal e-waste workers over time and further investigate if there are seasonal variations of PM lung function outcome parameters within e-waste worker group categories and across study sites.

The manuscript comprise six thematic areas; a brief abstract that summarizes the study, an introduction including the problem statement and objectives. The materials and methods section outlines the study area, the study participants and the methods used in the study. The results are presented using graphs and tables for further illustration followed by a comprehensive discussion of results including the strengths and limitations of the study. The manuscript conclusion summarizes the observations made, study limitations and future outlook.

## 2. Materials and Methods

### 2.1. Study Area

The Agbogbloshie informal e-waste recycling site (Figure 1) in Accra, is located in the central business district, bustling with several commercial and industrial activities, and a densely populated settlement [22]. The site is rated by Pure Earth among the top ten most contaminated areas on earth [23,24]. E-waste recycling and recovery of valuable parts include activities such as collecting, sorting, dismantling and open air burning of electrical cables to recover copper, each activity presenting unique occupational and environmental health risks. Of major public health concern is the open air burning of sheathed cables with fuel sources such as old discarded car tyres, and foams recovered from old refrigerators etc. [13] which results in the release of mixtures of several harmful substances, including, PM of varying sizes, polyaromantic hydrocarbons, and heavy transitional metals. The airborne particulates released in the burning process get suspended in immediate ambient environment and disperses into surrounding communities. Direct exposure of e-waste recyclers to these toxic chemicals including smoke could greatly increase workers’ risk for adverse respiratory impairment such as lung function changes, asthma, COPD (chronic obstructive pulmonary disease) and lung cancer.

Sixty-four (n = 64) male residents from Madina Zongo, a suburb of Accra in the La Nkwantanang sub-metropolitan district located more than 10 km from the Agbogbloshie e-waste site were selected as the control population. Madina Zongo was selected as a control population because of similarities in ethnicity and religion, as similar to e-waste workers at Agbogbloshie, majority are of the Dagomba tribe who migrated from northern Ghana, as well as being Muslims [25].

### 2.2. Study Design

We conducted a longitudinal cohort study comprising four rounds of repeated exposure and health outcome measurements among e-waste workers at Agbogbloshie and Madina Zongo from March 2017 to November 2018. After series of community entry procedures to introduce the study and obtain consent from leaders, study staff contacted the workers with the help of recruiters, read out and explained the participant information pamphlets and procedures. Workers who agreed to participate were recruited after they provided written consent by signature or thumbprint. This continued until a total of one hundred (100) male e- waste workers and fifty-one (51) male controls from Madina-Zongo were recruited. Originally, recruitment was planned at round 1 only, however, due to the high participant attrition of >30% at round 2, new participants were enrolled (42 and 14 for e-waste workers and controls respectively), to replace those that were lost to follow-up. Subsequently, there were three follow-up data measurements after round 1. Participants were located via mobile phone calls and the help of some identified e-waste workers and individuals in the two communities.

PM exposure concentration in participants breathing zone and lung function parameter measurements were collected from all participants at each round. Spirometry was performed by qualified medical personnel to measure the lung function of participants, and height and weight were also measured for body mass index estimation. A comprehensive respiratory health questionnaire was used to collect demographic information (age, gender, religion, ethnicity, and education, measures of socioeconomic position, location of birth and childhood and location of all residences), previous work and home inhalational exposures, and personal and family medical history through personal interviews. Data collection periods were aligned to the dry, rainy and harmattan (dry and dust-laden winds blowing from the Sahara desert into the Gulf of Guinea) [26] seasons to assess the effect of seasonality on PM levels and relationship with lung function. At the end of round 4, of the 207 participants recruited, 61 completed all four waves, 82 completed three waves, 25 completed two and 39 completed one wave. Informed consent was obtained, and questionnaires were administered by trained, interpreters in the participants’ preferred language of either Dagbani, Hausa, Twi, or English. The participants were compensated for their time and loss of daily wages (usually two working days). Each participant received 30 Ghana Cedis (approximately US$7), a branded T-shirt, and a snack meal. We obtained ethical approval (CHS-Et/M.4-P 3.9/2015–2016) from the Universities of Ghana and Michigan Institutional Review Boards (IRBs).

### 2.3. Particulate Matter Exposure Assessment

Comprehensive description of PM determination can be found in [27]. Briefly, PM in the breathing zone of participants was measured using a MetOne Aerocet 831 particle mass profiler (MetOne Instruments Incorporated, Grants Pass, OR, USA) which simultaneously measured PM2.5 and PM10 levels per minute at a flow rate of 2.8 L/min. This battery-operated sampler was incorporated into a backpack worn by participants whiles preforming their routine work tasks. Sampling was done for a 4-h period of the work shift. However, during the harmattan season, PM collection was done for 2 h as the Aerocet pump got overloaded quickly. Readings were recorded at beginning of each deployment and at retrieval respectively. The coarse fraction component was derived by estimating the difference in the concentrations of PM2.5 and PM10.

#### 2.3.1. Lung Function Measurement

Each participant performed lung function tests with the use of portable Easyone spirometer (ndd Medical Technologies, Andover, MA). Lung function test was performed by a qualified technician and physician according to the American Thoracic Society (ATS) guidelines [28]. Forced vital capacity (FVC), Forced expiratory volume in one second (FEV1), ratio of Forced expiratory volume in one second and Forced vital capacity (FEV1/FVC), peaked expiratory flow (PEF), and forced expiratory flow 25–75 (FEF25–75) and were measured in the standing position. All lung function data were expressed as predicted z-scores of the normal values according to the NHANES III equation [29]. A minimum of three acceptable FVC maneuvres were performed up to a maximum of six when reproducibility criteria were not met. An end of the test criterion of at least 2 seconds of no change in volume with an exhalation time of at least 6 seconds was used. The largest FVC and FEV1 obtained from any acceptable curves were used for recording purposes. A reproducibility criteria was peaked at less than 5% variation or 0.1L difference of the largest FVC and FEV1. Ethnically appropriate reference values were used for the interpretation of the results (Hankinson, Odencrantz, & Fedan, 1999) [30]. An abnormal test was defined as less than 80% or greater than 120% of predicted (percent predicted) for FVC and FEV1 and <70% for FEV1/FVC. The height and weight of the participants were also measured in the upright position without shoes and heavy clothing.

#### 2.3.2. Covariates

At enrollment, data on demographics (age, sex, religion, ethnicity, education, occupation, socioeconomic and marital status), detailed current and previous job history and exposures, home exposures including indoor cooking and biomass fuel use, pre-existing medical conditions, and lifestyle habits such as cigarette smoking and alcohol use were collected using a comprehensive questionnaire. Data on cigarette smoking included age at smoking initiation, duration of smoking, sticks smoked per day, current or ex-smoker defined as smoking cessation greater than a month’s duration. Smokers were classified as current, ex-, or never–smokers. The questionnaire was administered by trained interviewers in either Dagbani, Twi or English languages according to the participant’s preference. Limited interview guides were administered in subsequent follow-up data collection that explored possible changes in home and work exposures, and other respiratory conditions.

### 2.4. Statistical Procedures

Descriptive data were expressed as means and standard deviation, frequency, and percentages where appropriate. A bivariate analysis was performed to compare characteristics of study participants by location using student t-test statistics, and Spearman’s rank correlation coefficient and Chi-square test of independence to test association between e-waste workers and controls. All the lung function indices were log-transformed to approximate normality. The effects of PM (2.5, 2.5–10 and 10 µm) exposures on lung function parameters, (FVC, FEV1, FEV1/FVC ratio PEF, and FEF25–75) parameters were investigated by fitting a random-effects model with random slope effects terms. We used the model to examine the effect of PM (2.5, 2.5–10 and 10 µm) on lung function parameters, while controlling for age, education, job category, height, cigarette smoking status, location and seasonal variation. The regression coefficients and confidence intervals were documented. The lung function parameters were presented as Z-score in lung function per 10 µg/m³ concentration change in the fine and coarse PM in the participants breathing zones.

For the direct effect of job category on the respiratory outcomes, a power of 90% was used. This power was computed, using [31]’s formulas for longitudinal studies, at a significance level of 0.05, an exchangeable covariance structure, taking most set parameters (estimated FEV1% for controls of 90 and SD at 12.5) as those observed in previous similar studies in northern part of Ghana from where participants migrated to Accra.

## 3. Results

### 3.1. Socio-Demographic Characteristics of Study Participants

Generally, e-waste workers were younger (26.45 ± 6.74) compared to the controls (30.97 ± 9.92) [Table 1]. More than half, (51.94%) of study participants were within the 21–30 year age category. 77 (56%) of e-waste workers had no formal or primary education compared to the control population with 61% (46) of respondents having attained a high school education. Mean age, daily income, and educational level were significantly lower among the e-waste workers than the control population. The study found no significant differences in BMI due to exposure to particulate matter and lung function among e-waste workers and non-e-waste workers (*p* = 0.100).

The frequency of cigarette smoking among the e-waste workers was 32.65% (45) compared to 10% (6) among the non- e-waste workers. The association was statistically significant (*p* < 0.001). No significant differences were observed for daily income, alcohol use and previous history of pulmonary tuberculosis among the study participants. However, there was a higher prevalence of self- reported past history of bronchial asthma among the e-waste workers compared to the control group [3.55% and 3.33%/(*p* < 0.01)].

### 3.2. Seasonal and Site-Specific Variations of Particulate Matter in the Breathing Zone of Participants

Our results show that PM concentrations varied across the seasons with the highest concentrations recorded during the harmattan season. Generally, the median/IQR concentrations of PM among the e-waste workers at Agbogbloshie and the control population across the dry, rainy and harmattan seasons were consistently higher than the WHO air quality standards of 25 μg/m^3^ and 50 μg/m^3^ per 24-h mean. The observed median PM_2.5_ concentration levels at the e-waste and control sites were 69.86 ± 36.33, 61.18 ± 37.92 and 70.69 ± 4825 μg/m^3^ during the dry, rainy and harmattan seasons. In comparison, the concentrations for the control population were 34.88 ± 0.14.72, 34.13 ± 7.22 and 50.21 ± 169.64 across the three seasons respectively (*p* < 0.001, *p* < 0.001 and *p* = 0.950) [Appendix A]. PM2.5 concentrations were 2–3x times higher among the e-waste workers compared to the control group. The highest concentration of PM_2.5_ were recorded during the harmattan season with a significant association observed during the dry and rainy seasons (*p* < 0.001). The mean levels of PM_2.5_ among the e-waste workers were approximately twice that of the control population (*p* < 0.001). Similar trends were observed for the PM2.5–10 fraction across the dry, rainy and harmattan seasons [94.26 ± 87.34 μg/m^3^ vs. 68.23 ± 63.31 μg/m^3^; *p* = 0.009]; [48.88 ± 84.68 μg/m^3^ vs. 34.74 ± 55.41 μg/m^3^; *p* = 0.009] and [54.30 ± 126.38 μg/m^3^ vs. 31.59 ± 383.81 μg/m^3^; *p* = 0.960] respectively. Concentration of PM10 during the three seasons were 214.43 ± 154.46 μg/m^3^, 173.49 ± 96.12 μg/m^3^ and 181.64 ± 104.46 μg/m^3^ among the e-waste workers [118.12 ± 79.99 μg/m^3^/*p* ≤ 0.001, 99.35 ± 107.1 μg/m^3^/*p* ≤ 0.001 and 395.57 ± 412/*p* = 0.998] among the control participants.

Further analysis was conducted to determine the variation of PM in the breathing zone of the e-waste work- specific job categories across the sampling period and compared to the control population [Appendix A]. Generally, the median/IQR concentration of PM (2.5, 2.5–10 and 10µm) in the breathing zone of the e-waste worker categories were consistently higher across the seasons and the associations were statistically significant. Moreover, we consistently observed the highest concentrations of PM (2.5, 2.5–10 and 10 µm) among the e-waste burners across the seasons compared to the non-burners (dismantlers, sorters and collectors). High concentrations of PM (2.5, 2.5–10 and 10 µm) among collectors could be attributed to PM exposure from vehicular traffic as collectors trek around scavenging for wasted electrical gadgets for recycling.

### 3.3. Measurement of Lung Function Parameters among Study Participants

The mean (±sd) of the lung function of participants across the waves have been captured in [Appendix A]. Generally, lung function parameters were significantly lower among e-waste workers than controls. FEV1 and FVC ranged from [2.83 ± 0.54L/s, 3.84 ± 1.02L] and [2.95 ± 0.54L, 3.85 ± 1.23L] respectively among e-waste workers and the control group across the three seasons. The mean FEV1and FVC for the control group was slightly higher than the e-waste workers, however, the association was not statistically significant. The FEV1/FVC ratio were however lower among the e-waste workers compared to the control group (*p* ≤ 0.001, *p* = 0.007 and *p* = 0.007) across the dry, rainy and harmattan seasons respectively. Comparing the lung function parameters (FEV1, FVC, PEF and FEV1/FVC) among the e-waste and the non-waste workers the boxplot [Appendix A]. Generally, within site, FVC and FEV1 did not significantly vary across the seasons. It was observed that there is a greater variability as well as larger outliers for Agbogbloshie compared to Madina. At 95% confidence interval, the true medians for FEV1 and FVC did not differ significantly across the sites. Conversely, PEF and FEV1/FVC showed greater variability for Madina- Zongo participants compared to Agbogbloshie. Though the true medians appears not to differ at both sites for PEF and FEV1/FVC, that of Madina was observed to be higher compared to participants at the Agbogbloshie site.

### 3.4. Effect of Personal PM Exposure on Lung Function

Results show that PM (2.5, 2.5–10 and 10 µm) exposure affected the lung function parameters of the e-waste workers although the change in the effect sizes were small [Appendix A]. The results of the study showed that PM (2.5, 2.5–10 and 10 µm) in personal air was not significantly associated with lung function percentage of predicted at a 95% confidence level. The association was not statistically significant even after adjusting for potential confounders including seasons, job category, smoking status, past history of asthma, age, BMI and height [Table 2].

We further assessed the association between lung function parameters and the effects of the potential confounding factors [Appendix A]. Seasonality significantly influenced all the lung function parameters. It was observed that during the rainy season, FEV1 decreased by 1.3% [β = −0.013; 95% CI: −0.124–0.098], FVC by 8.2% [β = −0.082; 95% CI: −0.014–0.177], FEV1/FVC ratio decreased by 5.9% [β = −0.033; 95% CI: −0.87–0.021]], PEF by 12.5% [β = −3.133; 95% CI: −0.243, −0.022], and FEF25-75 by 17.9 [β = −0.266; 95% CI: −0.437, 0.094]. However, there were no significant differences at 5% level of significance. Greater decreases were also observed during the harmattan season, (FEV1 −7.5% [β = −0.075; 95% CI: −0.165–0.051], FVC −2.4% [β = −0.024; 95% CI: −0.102–0.052], FEV1/FVC ratio −3.3% [β = −0.033; 95% CI: −0.87–0.021], PEF −13.3% % [β = −3.133; 95% CI: −0.243, −0.022] and FEF 25–75 by 26.6% [β = −0.266; 95% CI: −0.437, 0.094]. Harmattan season showed significant influence especially on PEF and FEF 25–75.

With respect to job category, e-waste burners showed the highest decreases in their lung function parameters compared to the other job categories. The risk of work as a burner was predicted to decreased FEV1 by 4.5% [β = −0.045; 95% CI: −0.151–0.061], FEV1/FVC by 6.0% [β = −0.060; 95% CI: −0.123–0.004], PEF by 14.2% among e-waste burners [β = −0.142; 95% CI: −0.278, −0.008] and FEF_25–75_ by 22.5% [β = −0.225; 95% CI: −0.451–0.001]. Past history of Asthma was also significantly associated with the lung function parameters. Specifically, participants with a previous history of Asthma were predicted to have decreased FEV1 by 35.8 % [β = −0.358; 95% CI: −0.590, 0.125] [Appendix A].

## 4. Discussion

Our study examined the effects of PM (2.5, 2.5–10 and 10 µm) exposure in personal air on lung function among e-waste workers at Agbogbloshie, an informal recycling site and a control population from Madina-Zongo, in Accra Ghana. PM concentrations were consistently 2–3 times higher among the e-waste workers compared to the control group. The median/IQR values of PM (2.5, 2.5–10 and 10 µm) concentrations across the three seasons at both sites exceeded the WHO air quality standards of 25 μg/m^3^ and 50 μg/m^3^ per 24–hour mean and the average in-country ambient PM_2.5_ concentration levels of 35 µg/m [32]. [33], also reported significantly high concentrations of PM (2.5, 2.5–10 and 10 µm) in the breathing zone of individuals who were involved in consistent burning of e-waste in Thailand. They reported mean PM2.5 and PM 2.5–10 concentrations of 441 µg/m^3^ and 2274 µg/m³ respectively.

Lung function measurement is an objective measure of respiratory health and a predictor of cardiopulmonary morbidity and mortality [6]. The larger airway parameters of lung function test such as FVC, FEV1, and FVC/FEV1 measure lung capacity and large airway resistance and have been commonly used in previous occupational studies to assess the relationship between particulate matter air pollution and pulmonary function [9,10,34,35,36,37,38]. Although the effect sizes for assessing the association of PM exposure on lung function parameters were small, it does not eliminate the potential adverse effects on lung function of the workers. In fact, according to world Health Organization (WHO, 2005) [39], there is no evidence of a safe threshold below which no adverse health effects of PM may occur. We postulate that competent immune status, age and the healthy worker effect may have contributed to the small and statistically insignificant effect observed in the e-waste workers. Consistent with our findings, previous studies conducted in occupational settings with high exposures to PM (2.5, 2.5–10 and 10 µm) [9,40,41,42] reported significant decline in lung function parameters (FEV1, FVC, FEV1/FVC), among the study participants due to exposures to PM2.5, PM 2.5–10 and PM10. Similarly, our study found a significant risk of lung function decline among e-waste workers, from exposure to high concentration of PM. To the best of our knowledge there has been no published studies examining the associated effects of PM exposure on lung function among e-waste workers. However, Amoabeng Nti, (2015) in a master’s thesis, assessed the effect of heavy metals in blood on the lung function of 20 e-waste burners from Agbogbloshie and reported that Pb in blood reduced FVC by 0.20 L, Cd in urine reduced FVC by 0.143 L, Ni in urine reduced FVC by 0.294 L [43]. However, none of the associations between the heavy metals and FVC were significant. Also, two cross-sectional studies conducted at informal e-waste recycling sites in China and examined the association between exposure to heavy metals in particulates and lung function among children living near e-waste dumpsite in Guiyi [44,45]. We observed a consistent decline in FEV1 percentage of predicted among all the e-waste worker categories, more among burners than dismantlers, after adjusting for seasonal variation, age, and cigarette smoking. The effects of cigarette smoking, job category and indoor cooking, among e-waste workers were higher risk factors for reduced lung function. Our study showed a negative association of cigarette smoking on the lung function parameters. This is in contrast with findings from other occupational studies where workers were exposed to PM-related pollutants. Szram et al., (2012) conducted a systematic review and meta-analysis of welding fumes exposure on lung function among welders and reported −9.0 mL/year^−1^ (95% CI −22.5–4.5; *p* = 50.193) in FEV1 decline between welders and non-welders and an annual FEV1 decline between welders and controls with smoking history of −13.7 mL/year^−1^ (95% CI −33.6–6.3; *p* = 50.179) [44]. Bolund and colleagues 2015) also conducted a longitudinal study among young farmers and reported a negative effect of [−0.12, *p* = 0.006] and [–0.15, *p* = 0.009] on FEV1, FEV1/FVC and FVC respectively [43] and that current smoking had a deleterious effect on lung function. Our study findings further corroborate previous studies that continuous exposure to PM2.5, PM 2.5–10 and PM10 may predispose to increased risk for lung function decline and chronic respiratory disease [2,9,12,19].

Lung function measurement is an objective measure of respiratory health and a predictor of cardiopulmonary morbidity and mortality [6]. The larger airway parameters of lung function test such as FVC, FEV1, and FVC/FEV1 measure lung capacity and large airway resistance and have been commonly used in previous occupational studies to assess the relationship between particulate matter air pollution and pulmonary function [9,10,34,35,36,37,38]. Although the effect sizes for assessing the association of PM exposure on lung function parameters were small, it does not eliminate the potential adverse effects on lung function of the workers. In fact, according to world Health Organization (WHO, 2005) [39], there is no evidence of a safe threshold below which no adverse health effects of PM may occur. We postulate that competent immune status, age and the healthy worker effect may have contributed to the small and statistically insignificant effect observed in the e-waste workers. Consistent with our findings, previous studies conducted in occupational settings with high exposures to PM (2.5, 2.5–10 and 10 µm) [9,40,41,42] reported significant decline in lung function parameters (FEV1, FVC, FEV1/FVC), among the study participants due to exposures to PM2.5, PM 2.5–10 and PM10. Similarly, our study found a significant risk of lung function decline among e-waste workers, from exposure to high concentration of PM. To the best of our knowledge there has been no published studies examining the associated effects of PM exposure on lung function among e-waste workers. However, Amoabeng Nti, (2015) in a master’s thesis, assessed the effect of heavy metals in blood on the lung function of 20 e-waste burners from Agbogbloshie and reported that Pb in blood reduced FVC by 0.20L, Cd in urine reduced FVC by 0.143L, Ni in urine reduced FVC by 0.294L [43]. However, none of the associations between the heavy metals and FVC were significant. Also, two cross-sectional studies conducted at informal e-waste recycling sites in China and examined the association between exposure to heavy metals in particulates and lung function among children living near e-waste dumpsite in Guiyi [44,45]. We observed a consistent decline in FEV1 percentage of predicted among all the e-waste worker categories, more among burners than dismantlers, after adjusting for seasonal variation, age, and cigarette smoking. The effects of cigarette smoking, job category and indoor cooking, among e-waste workers were higher risk factors for reduced lung function. Our study showed a negative association of cigarette smoking on the lung function parameters. This is in contrast with findings from other occupational studies where workers were exposed to PM-related pollutants. Szram et al., (2012) conducted a systematic review and meta-analysis of welding fumes exposure on lung function among welders and reported −9.0 mL/year-^1^ (95% CI −22.5–4.5; *p* = 50.193) in FEV1 decline between welders and non-welders and an annual FEV1 decline between welders and controls with smoking history of −13.7 mL/year- (95% CI −33.6–6.3; *p* = 50.179) [44]. Bolund and colleagues 2015) also conducted a longitudinal study among young farmers and reported a negative effect of [−0.12, *p* = 0.006] and [−0.15, *p* = 0.009] on FEV1, FEV1/FVC and FVC respectively [43] and that current smoking had a deleterious effect on lung function. Our study findings further corroborate previous studies that continuous exposure to PM2.5, PM 2.5–10 and PM10 may predispose to increased risk for lung function decline and chronic respiratory disease [2,9,12,19].

Furthermore, the higher increases in the reduction of smaller airway parameters (PEF and FEF25-75%) suggest that the e-waste workers may be at an increased risk for small airway diseases such as occupational asthma, bronchitis, bronchiolitis and COPD due to airway obstruction. Small airways (bronchioles) appear particularly vulnerable to obstructive diseases because of their narrow lumen. Thus, many particulates may be deposited there making them more susceptible to complete obstruction than larger airways.

There are other potential sources of inhalational exposures that may affect the respiratory function of the informal e-waste workers. These include gaseous (e.g. nitrogen dioxide and ozone), organic (e.g., PAHs) and inorganic (heavy metals and vehicular exhaust emissions), other pollutants from e-waste recycling activities, refuse burning at nearby landfill site and combustion emissions from various informal local restaurants (chop bars) dotted around the recycling site. However, due to financial constraints, their effects were not evaluated in this study.

Our study had some important strengths. To the best of our knowledge, this is the first longitudinal cohort study with repeated measures to investigate the association between recurrent exposure to PM2.5, PM 2.5–10 and PM10 in the breathing zone and lung function parameters of informal e-waste workers.

Important limitations to our study must be noted. Firstly, the observed effects may be biased by the assumption of the exposure time and lung function measurement. The 4-h mean PM concentration on the days of spirometry measurements, was assumed to be suitable for this study. Secondly, we realize that the best estimates require the use of monitoring devices that can be carried by e-waste workers over a longer period than four working hours. Thirdly, the evaluation of the effects of air pollutants by season would further have been strengthened if the interaction between gaseous pollutants such as ozone, nitrogen dioxide and UFP PM were considered.

Fourthly, other particulates can also occur in both gaseous and liquid states, however, the concentrations of PM were evaluated only by mass concentration. There is the need for further evaluation to include the various states of matter. Finally, indoor air pollution should be considered and objectively determined in future studies. Indoor biomass fuel use was considered as a confounding factor, however, an objective measurement for direct effect assessment would have been ideal. Further research is needed to examine the effects of PM pollution on e-waste workers’ lung function and PM concentrations exposure.

## 5. Conclusions

Exposure to elevated levels of PM in the breathing zone is a risk factor for decreased lung function parameters among healthy e-waste workers resulting in reduced spirometry results. The negative effects of PM (2.5, 2.5–10 and 10 µm) on the lung function of healthy e-waste workers requires an effective workplace ambient air quality management program, necessary to reduce the workers’ respiratory health risks. There is the need for improvement in the work layout, methods and procedures with reduced exposure to hazardous agents injurious to the lungs of e-waste workers. Health and safety promotion campaigns should be conducted emphasizing the hazards of exposure to fine and coarse PM to mitigate these health effects. This study however, was limited by shorter duration of personal PM measurement as well as shorter intervals for subsequent follow-up sampling periods. A longer duration of sampling would have further strengthened the assessment of changes in lung function from PM exposures. Being a biological phenomenon, the bodies coping mechanisms and healthy worker effects may delay any observable clinical adverse effects.

## Figures and Tables

**Figure 1 ijerph-17-03042-f001:**
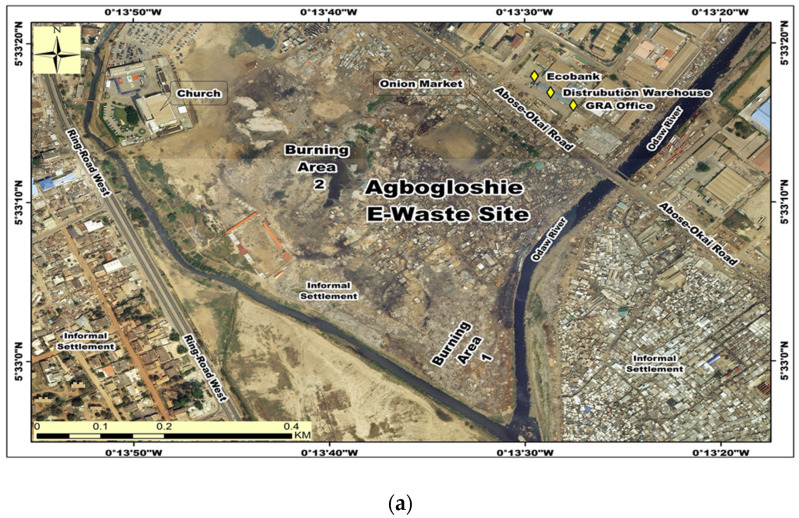
Google earth map views of study sites, Agbogbloshie and Madina-Zongo, Accra-Ghana. (**a**) e-waste recycling (exposed site); (**b**) Madina- Zongo (Control site).

**Table 1 ijerph-17-03042-t001:** Sociodemographic characteristics of study participants. Bold *p*-values represent significant values (*p* ≤ 0.05).

Variable	Total (N = 207)	E-Waste Workers (N = 142)	Non E-Waste Workers (N = 65)	*p*-Value
	n (%)	n (%)	n (%)	
**BMI (kg/m^2^): mean ± SD**	23.37 ± 3.19	23.19 ± 3.05	23.75 ± 3.44	0.100
**SPO2** (mean ± SD)	96.56 ± 3.23	96.34 ± 3.64	97.06 ± 1.94	**0.019**
**Age in years: mean ± SD**	27.87 ± 8.14	26.45 ± 6.74	30.97 ± 9.92	**<0.001**
**Age category**				**0.05**
**≤20 years**	39(18.93)	30 (21.28)	9 (13.85)	
**21–30 years**	107(51.94)	77(54.61)	30(46.15)	
**31–40 years**	41(19.9)	27(19.15)	14(46.15)	
**≥40 years**	19(9.22)	7(4.96)	12(18.46)	
**Education**				**<0.001**
None	45(22.96)	39(27.86)	6(10.71)	
Primary	42(21.43)	38(27.14))	4(7.14)	
Middle/JHS	61(31.12)	44(31.43)	17(30.36)	
Secondary/SHS+	48(24.49)	19(12.86)	29(31.29)	
**Marital Status**				**0.05**
Married	103(50.74)	82(58.16)	21(33.87)	
**Daily Income**				**<0.001**
≤GHS 20	36(18.00)	26(18.44)	10(16.95)	
GHS 20–40	59(29.5)	46(32.62)	13(22.03)	
GHS 41–60	57(28.5)	38(26.95)	19(32.2)	
GHS 61+	48(24.0)	31(20.91)	16(3.28)	
**Living arrangements**				
Sleep location after work				**<0.001**
On the site	59(42.45)	59(42.45)	-	
Off site, but within 1km of Agbogbloshie	66(34.20)	66(47.48)	-	
More than 1 km away		-	65(100.0)	
**Indoor cooking**				**<0.001**
Yes	45(22.73)	22(15.94)	23(38.33)	
**Lifestyle habits**				
**Alcohol use**				0.181
Current	24(11.94)	19(13.48)	5(8.33)	
Previous	7(3.48)	3(2.13)	4(6.67)	
Never	170(84.58)	119(84.40)	51(85.00)	
**Cigarette smoking**				**0.001**
Current	50(25.13)	45(32.65)	6(10)	
Never	149(74.87)	95(68.35)	54(90)	
**Medical History**				
Pulmonary Tuberculosis				0.86
Yes	2(1)	2(1.42)	0(0)	
Asthma				**0.010**
Yes	7(3.48)	5(3.55)	2(3.33)	

**Table 2 ijerph-17-03042-t002:** Percent change (%) and 95%CI in lung function parameters per interquartile change range (IQR) of the respective PM (results from random effects linear regression models, *p* ≤ 0.05).

	FEV1		FVC		FEV1/FVC		PEF		FEF25–75	
PM Fraction	*β* [95% CI]	*p*-Value	*β* [95% CI]	*p*-Value	*β* [95% CI]	*p*-Value	*β* [95% CI]	*p*-Value	*β* [95% CI]	*p*-Value
**PM2.5**	−0.001(−0.005,0.004)	0.777	−0.001(−0.006,0.003)	0.531	−0.001(−0.004,0.002)	0.702	0.006(−0.001,0.012)	0.098	0.005(−0.002,0.012)	0.181
**PM10**	−0.001(−0.006,0.004)	0.651	−0.002(−0.007,0.003)	0.394	−0.001(−0.004,0.003)	0.773	0.007(−0.002,0.017)	0.121	0.012(0.003,0.021)	**0.013**
**PM2.5–10**	0.002(−0.006,0.009)	0.630	0.003(−0.004, 0.010)	0.390	0.001(−0.004,0.005)	0.842	−0.008(−0.020,0.003)	0.163	−0.015(−0.026,−0.003)	**0.010**

* Adjusted for Age, BMI, indoor cooking, seasonal variation, cigarette smoking, and job category Abbreviations: FEV1—Forced expiratory volume in one second; FVC—forced vital capacity; FEV1/FVC—ratio of Forced expiratory volume in one second and forced vital capacity; PEF—Peak vital capacity; FEF25-75—forced expiratory flow 25–75; FEV1/FVC ratio ≤ 0.7. *β*—Coefficient, CI: confidence interval. Bolden p-values are statistically significant.

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
