# Peer review of "Effect of Particulate Matter Exposure on Respiratory Health of e-Waste Workers at Agbogbloshie, Accra, Ghana"

_ijerph, 2020, doi:10.3390/ijerph17093042_

Round 1

Reviewer 1 Report

Dear authors,

Below are some comments and questions that need to be addressed to strengthen your manuscript.

  1. Introduction

Line 65: change augment to impact

Line 74: What do you mean by “ecological media”

Line 92: change to dyspnea

Line 94: is either airborne particles or PM, having both terms together is redundant.

  1. Materials and Methods

There are two subsections numbered 2.2 (Line 122 and Line 128).

Statistical procedures: Did you perform analyses with z-scores instead of % predicted?  What about assessing the lag effects? In line 209, why per 1µg/m3 change instead of 10µg/m3?  Did you perform a power analysis for the study?

  1. Results

This section is confusing with the numbering of the tables and supplemental information.  The figures/tables need better descriptions, the axes and legends must be improved. I am assuming that in the text the numbers in parenthesis means n=#, please specify.  Also, some sentences are not complete or have typos (see line 221-224 for an example). What is the harmattan season (line 239), in the methods only a dry and a wet season are described.  Add graph with the comparisons of PM concentrations among seasons as Table 2 – S1 only shows the comparison between the exposure and control sites for each season.  For figure 2, please specify box plot parameters and the n for each round.  For tables in S4 and S5, is the data for all the subjects or just the e-waste workers? 

  1. Discussion

Please discuss what other potential exposures the workers have at the facilities that might also impact lung function.  It is important to address in the limitations that other air pollutants that could impact the results were not measured, even though in the introduction it is mention that open air burning is a common practice during e-waste recycling. Discuss why the time variable was not assessed even though the study was longitudinal in nature. What is the power of the study?

Author Response

Dear Reviewer,

We are grateful for your comments that have improved the overall content of our manuscript. Kindly find attached our responses to your comments.

Thank you.

Kind regards,

Afua Asabea Amoabeng Nti

Reviewer 2 Report

This is an interesting study. There is a conflicting issue of high incidence of smoking among e-waste workers compared to control population. Considering the data is at hand, how do the lung function parameters look like if smokers and non-smokers are in separate categories.

Author Response

Dear Reviewer,

We are grateful for your comments that have improved the overall content of our manuscript. Kindly find attached our responses to your comments.

Thank you.

Sincerely

Afua Asabea Amoabeng Nti

Reviewer 3 Report

L97 - I suggest to authors insert a brief paragraph about the manuscript structure. 

L344-352 - I suggest to authors discuss about the limitations of the work done in Conclusion section.

Author Response

Dear Reviewer,

We are grateful for your comment that have improved the overall content of our manuscript. Kindly find attached our responses to your comments.

Thank you.

Sincerely,

Afua Asabea Amoabeng Nti

Round 2

Reviewer 2 Report

Thanks for addressing my concern.